# Traumatic Brain Injury Characteristics Predictive of Subsequent Sleep-Wake Disturbances in Pediatric Patients

**DOI:** 10.3390/biology11040600

**Published:** 2022-04-14

**Authors:** Brittany Gerald, J. Bryce Ortiz, Tabitha R. F. Green, S. Danielle Brown, P. David Adelson, Sean M. Murphy, Rachel K. Rowe

**Affiliations:** 1Department of Child Health, University of Arizona College of Medicine-Phoenix, Phoenix, AZ 85004, USA; brittanymgerald@email.arizona.edu (B.G.); jbryceortiz@arizona.edu (J.B.O.); tabithagreen@email.arizona.edu (T.R.F.G.); dadelson@phoenixchildrens.com (P.D.A.); smmurp2@arizona.edu (S.M.M.); 2Barrow Neurological Institute, Phoenix Children’s Hospital, Phoenix, AZ 85016, USA; dbrown4@phoenixchildrens.com; 3Phoenix VA Health Care System, Phoenix, AZ 85012, USA; 4Department of Integrative Physiology, University of Colorado, Boulder, CO 80301, USA

**Keywords:** hypersomnia, concussion, adolescence, traumatic brain injury, circadian rhythm, insomnia

## Abstract

**Simple Summary:**

Traumatic brain injury is a leading cause of death and disabilities in children and adolescents. Poor sleep after brain injury can slow recovery and worsen outcomes. We investigated clinical sleep problems following pediatric brain injury. We examined characteristics of the injury and details about the patients that may be risk factors for developing sleep problems. The number of patients that developed problems with their sleep after a brain injury was similar between genders. The probability of insomnia increased with increasing patient age. The probability of ‘difficulty sleeping’ was highest in 7–9 year-old brain-injured patients. Older patients had a shorter time between brain injury and sleep problems compared to younger patients. Patients with severe brain injury had the shortest time between brain injury and development of sleep problems, whereas patients with mild or moderate brain injury had comparable times between brain injury and the onset of poor sleep. Multiple characteristics of brain injury and patient details were identified as risk factors for developing sleep problems following a brain injury in children. Untreated sleep problems after a brain injury can worsen symptoms, lengthen hospital stays, and delay return to school. Identifying risk factors could improve the diagnosis, management, and treatment of sleep problems in survivors of pediatric brain injury.

**Abstract:**

The objective of this study was to determine the prevalence of sleep-wake disturbances (SWD) following pediatric traumatic brain injury (TBI), and to examine characteristics of TBI and patient demographics that might be predictive of subsequent SWD development. This single-institution retrospective study included patients diagnosed with a TBI during 2008–2019 who also had a subsequent diagnosis of an SWD. Data were collected using ICD-9/10 codes for 207 patients and included the following: age at initial TBI, gender, TBI severity, number of TBIs diagnosed prior to SWD diagnosis, type of SWD, and time from initial TBI to SWD diagnosis. Multinomial logit and negative-binomial models were fit to investigate whether the multiple types of SWD and the time to onset of SWD following TBI could be predicted by patient variables. Distributions of SWD diagnosed after TBI were similar between genders. The probability of insomnia increased with increasing patient age. The probability of ‘difficulty sleeping’ was highest in 7–9 year-old TBI patients. Older TBI patients had shorter time to SWD onset than younger patients. Patients with severe TBI had the shortest time to SWD onset, whereas patients with mild or moderate TBI had comparable times to SWD onset. Multiple TBI characteristics and patient demographics were predictive of a subsequent SWD diagnosis in the pediatric population. This is an important step toward increasing education among providers, parents, and patients about the risk of developing SWD following TBI.

## 1. Introduction

Traumatic brain injuries (TBIs) in the pediatric population can lead to debilitating and lifelong morbidities that include disruption to cognitive [1,2,3], physical, developmental, psychological [4,5], and endocrine processes [6,7,8]. Moreover, recent literature has revealed that sleep-wake disturbances (SWD) are common after pediatric TBI [9,10,11,12,13], with several studies showing that disturbed sleep is one of the most frequently reported post-injury symptoms by pediatric TBI survivors and their caregivers [14,15]. Proper sleep is an important facet of health for all ages but is especially critical in children because of its role in brain development and maturation [16,17,18,19]. Notably, sleep disturbances can impair recovery from TBI [20,21,22]. As such, understanding how TBI impacts sleep in the pediatric population could lead to the improved treatment of children post-TBI.

The severity of TBIs is often defined by the patient’s neurological status at time of service using the Glasgow Coma Scale (GCS) [23], wherein TBIs range from mild (GCS 13–15), moderate (GCS 9–12), and severe (GCS 3–8). Although commonly used, the GCS has limitations due to the varying nature of presenting symptoms for TBI. For example, the presence of intracranial hemorrhage has a high risk of expansion and neurological decline despite an initial GCS of a mild or moderate rank [24]. Nevertheless, classifying TBI severity remains important because doing so informs patient care, which includes inpatient treatment, rehabilitation, discharge plans, and symptom resolution [23,25]. Several studies have shown that brain injury severity affects neurobehavioral outcomes during the months following injury. Children who experienced a moderate or severe injury exhibited poorer performance on higher-order cognitive tasks and impaired academic skills compared to children with mild injury severity [26,27]. In this study, TBI severity was determined using ICD-9/10-CM short diagnosis descriptions which were assigned, in part, by GCS scores.

Although the impacts of injury severity on SWD following pediatric TBI is understudied, clinical data indicate that adolescents who sustain moderate-to-severe TBI exhibit increased daytime somnolence compared to adolescents with mild TBI [28]. In this study, we aimed to understand the prevalence of SWD following pediatric TBI and how severity of TBI affected this outcome. We present a retrospective review and analysis of clinical data to understand the prevalence of post-TBI SWD in a pediatric population at a major United States metropolitan Level I pediatric trauma center, with a focus on characteristics of TBI and patient demographics that might be predictive of SWD development. We focused on the extent to which SWD occur between genders, across injury severities, and the time between TBI and SWD diagnoses. We also investigated whether the frequency or type of SWD increased with age at initial TBI, severity of TBI (mild, moderate, or severe), or in the setting of multiple TBIs. We hypothesized that pediatric TBI survivors would display subsequent SWD that could, in part, be predicted by TBI severity and the number of TBIs incurred.

## 2. Materials and Methods

### 2.1. Patient Search Queries

Following approval from the Phoenix Children’s Hospital Institutional Review Board (IRB-19-377), we performed a single-institution, retrospective chart review of a cohort of pediatric patients who presented to and were treated at Phoenix Children’s Hospital between 2008 and 2019. *International Classification of Diseases, 9th Revision, Clinical Modification* (ICD-9-CM), and the *International Classification of Diseases, 10th Revision, Clinical Modification* (ICD-10-CM) codes were used to search for and identify patients with a TBI and subsequent SWD diagnoses within two years of the TBI diagnosis (Appendix A). The transition from ICD-9-CM to ICD-10-CM codes occurred on 15 October 2015; ICD-10-CM was intended to update and expand on ICD-9-CM codes to keep up with new and evolving medical diagnoses. The description of the codes for TBI and SWD remained similar between the two coding systems but with significant changes to the code numbers. As the transition period was during the time span of our study, both ICD-9-CM and ICD-10-CM codes were queried.

### 2.2. Inclusion and Exclusion Criteria

At Phoenix Children’s Hospital between 1 January 2008 and 31 December 2019, there were 4822 admissions (2936 M:1886 F) with a TBI diagnosis. Patients were evaluated for adherence to the following inclusion and exclusion criteria. All pediatric patients, <18 years of age, who were treated at Phoenix Children’s Hospital between the years 2008 to 2019, were diagnosed with a TBI, and were subsequently diagnosed with an SWD within two years were included in this study. Patients treated for TBI that had a prior medical diagnosis of an SWD were excluded from this study. This resulted in a total of 207 patients who met the inclusion criteria and were retained for analysis. This is substantially lower than the published longitudinal studies in pediatric patients that report 39% of children experience poor sleep quality at 1-month post-injury, and 28% at 12 months post-injury [29]. Our inclusion criteria only captured patients seeking follow-up medical treatment at a single institution, in a hospital setting, which likely resulted in underreporting.

### 2.3. Patient Data

For each patient, we compiled data for the following variables: patient age at initial TBI, patient gender, severity of TBI, total number of TBIs diagnosed prior to SWD diagnosis, type of SWD, and we derived the time from initial TBI to the SWD diagnosis. For patients with multiple TBIs, time was calculated starting from the first diagnosed TBI. TBI severity was categorized based on ICD-9/10-CM short diagnosis descriptions. Mild TBI included concussions or TBI without loss of consciousness (LOC), moderate TBI included TBI with LOC but no imaging findings or neurological deficits, and severe TBI included TBI with LOC and imaging findings or neurological deficits. Based on the ICD-9/10-CM codes, types of SWD were reduced from 12 categories (see Appendix A) to the following six categories by combining similar categories that had small sample sizes (*n* < 10; [30]: (1) Circadian rhythm sleep disorder; (2) Hypersomnia and excessive daytime sleepiness; (3) Insomnia; (4) Obstructive sleep apnea (OSA); (5) Difficulty sleeping, difficulty falling asleep, disturbance in sleep, frequent awakening, and sleep difficulties and sleep disorders; and (6) Other. The six categories and corresponding sample sizes are provided in Table 1. The category ‘Other’ was assigned as a diagnosis at the time of treatment using the ICD-9/10-CM codes for ‘Other’ and is not a category created by the authors. Disorders were combined when group sizes were small and symptoms could be overlapping (e.g., hypersomnia and excessive daytime sleepiness). Category 5 was created by combining multiple diagnoses that had non-specific diagnostic criteria and small sample sizes.

### 2.4. Statistical Analyses

We conducted exploratory analyses to summarize the distributions of TBI, TBI severity, and SWD onset following TBI, relative to patient characteristics (e.g., gender and age). To accomplish this, violin plots and box plots were created using the *ggplot2* package in the R statistical computing environment [31,32]. We then fit multinomial logit models using the *nnet* package in R [33,34] to investigate if multiple types of SWD could be predicted by patient gender, age at initial TBI, severity of the initial TBI (mild, moderate, or severe), the total number of TBIs prior to the onset of SWD, and the total number of months from TBI to onset of SWD. We fit a total of 32 multinomial logit models that represented all possible singular and additive combinations of the predictor variables. We were also interested in potential factors that might predict the time between TBI and the onset of SWD. Time was recorded as the number of months, which was an overdispersed count variable ($\bar{x}$ = 4.02; σ^2^ = 35.16; dispersion = 8.03). Therefore, we fit negative-binomial generalized linear models using the *MASS* package in R [35,36]. We fit a total of 16 negative-binomial models that represented all possible singular and additive combinations of gender, age at initial TBI, severity of the initial TBI, and the total number of TBIs prior to the onset of SWD.

For both the multinomial logit and negative-binomial models, we implemented information-theoretic model selection via Akaike’s Information Criterion corrected for small sample size (AIC*_c_*) to identify the best predictive models [37,38]. We considered all models ≤2 ΔAIC*_c_* of the top-ranked model as competing [38]. However, we investigated all competing models for uninformative parameters, as per the methods described in [39]. If uninformative parameters were detected in competing models, then we only considered the top-ranked, most parsimonious model for obtaining estimates.

## 3. Results

Among the patients diagnosed with TBI at Phoenix Children’s Hospital during our study period, 207 were diagnosed with a subsequent SWD and met our inclusion/exclusion criteria. Although the gender ratio for those 207 patients was similar (52.7% M:47.3% F), their proportion relative to the total population of TBI patients was not (3.7% M:5.2% F). For example, a chi-square test indicated that males were more likely to be diagnosed with a TBI ($\chi_{1}^{2}$ = 228.64, *p* < 0.00001), but a two-tailed *z*-test for population proportions suggested that females who incur a TBI may be more likely to be diagnosed with an SWD than males who incur a TBI (*z* = 2.5, *p* = 0.01).

**Figure 1 biology-11-00600-f001:**
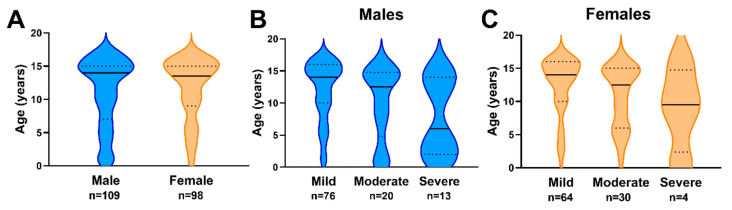
(**A**) Age distributions of patients by gender. (**B**) Age distributions of patients by TBI severity for males. (**C**) Age distributions of patients by TBI severity for females. Medians and quartiles are represented by solid black lines and dashed black lines, respectively.

The median age of males was 14 years, with lower and upper quartiles of 7 and 15 years, respectively (Figure 1A). The median age of females was 13.5 years, with lower and upper quartiles of 9 and 15 years, respectively (Figure 1A). Most patients with SWD were diagnosed with an initial TBI of mild severity in both males (69.7%; Figure 1B) and females (65.3%; Figure 1C), followed by moderate severity (males = 18.3%; females = 30.6%), and severe TBI (males = 11.9%; females = 4.1%), respectively (Figure 1B,C). The median number of TBIs prior to SWD for males was 2, with lower and upper quartiles of 1 and 3, respectively (Figure 2A). Females had a median of 1.5 TBIs prior to an SWD diagnosis, with a lower quartile of 1 and an upper quartile of 2 (Figure 2A). The median time to onset of SWD in males was 2 months, with lower and upper quartiles of 1 and 3 months, respectively (Figure 2B). The median time to onset of SWD in females was 1.5 months, with lower and upper quartiles of 1 and 2 months, respectively (Figure 2B).

Only one multinomial logit model was strongly supported (≤2 ΔAIC*_c_*), which included age at initial TBI as the single predictor (Table 2). This model estimated a positive effect of age on insomnia, such that the probability of insomnia increased with patient age when the first TBI occurred (Odds Ratio [OR] = 1.05; Figure 3C); patients who were ≥12 years of age (i.e., 12–17 years-old) when the first TBI occurred had the highest probability of insomnia (*p* > 0.20). Conversely, a negative effect of age on the category ‘other’ existed, such that the probability of other/uncategorized SWDs decreased with increasing patient age (OR = 0.81; Figure 3F); patients who were ≥12 years of age (i.e., 12–17 years-old) when the first TBI occurred had the lowest probability of the ‘other’ category of SWD (*p* ≤ 0.10). Interestingly, a concave effect of age on the difficulty sleeping category was estimated, such that the probability of difficulty sleeping was lowest at the youngest and oldest ages but was highest for 7–9 year-olds (OR = 0.93; Figure 3E). No support existed for effects of age on circadian rhythm sleep disorder (Figure 3A), hypersomnia (Figure 3B), or OSA (Figure 3D).

Two competing negative-binomial models were supported for predicting the time from initial TBI to the onset of SWD (Table 3). The top-ranked model included age at initial TBI, severity of the initial TBI, and the total number of TBIs prior to the onset of SWD as important predictors and are further described in Figure 4. The competing model (ΔAIC*_c_* = 1.67) included those same predictors with the addition of patient gender; however, the addition of gender resulted in one more model parameter but a nominal 0.23 change in log-likelihood relative to the nested top-ranked model, and the 95% confidence interval for the gender coefficient estimate considerably overlapped zero (95% CI = −0.42–0.20). Collectively, these findings indicated that gender was an uninformative variable; therefore, model-averaging was not conducted, and the top-ranked model was considered the most parsimonious (Arnold, 2010).

A strong negative effect of age was supported, whereby the incident rate of time to the onset of SWD decreased by 0.93% (95% CI = 0.64–1.32) for every one-year increase in age at initial TBI (Figure 4A). In contrast, a very strong positive effect of the total number of TBIs prior to SWD was supported, whereby the incident rate of time to onset of SWD increased by 1.19% (95% CI = 1.08–1.33) for every additional TBI that a patient incurred (Figure 4B). Additionally, a strong negative effect of injury severity was supported, whereby the incident rate of time to onset of SWD was 0.41× lower (95% CI = 0.21–0.81) for severe TBIs than for mild and moderate TBIs (Figure 4C).

## 4. Discussion

Previous studies have shown that children who incur TBI will likely have more sleep disturbances compared to children without TBI [40,41]. Our results provide evidence that SWD can occur after pediatric TBI, the prevalence of which might differ between genders. There was a similar distribution of SWD between genders and all statistical models that included gender as a predictor indicated that it was an uninformative variable. However, relative to the total population of TBI patients, females may be more likely to be diagnosed with an SWD than males. This result is similar to the findings of a comparable study that examined the prevalence and trajectory of sleep disturbances in children up to 24 months after a TBI, which found that female gender was a significant risk factor associated with increased sleep disturbances [42]. In contrast, other studies reported that TBI patients diagnosed with post-injury SWD, specifically fatigue, did not have differences in gender distributions compared with orthopedic controls [43]. Thus, additional studies are needed to evaluate gender as a possible risk factor for SWD after TBI in the pediatric population.

A recent prospective study that examined sleep disturbances after TBI reported that children and adolescents with moderate or severe TBI had significantly more difficulties with sleep compared to children with orthopedic injuries but no TBI [44]. Approximately 50% of pediatric TBI patients in that study had subclinical to clinical symptoms of insomnia [44]. Insomnia, the difficulty of initiating and maintaining sleep, can result in inadequate sleep and impact development in children. In the pediatric population, insomnia is one of the most frequently diagnosed SWD [45], specifically in adolescents [46]. We found that the probability of insomnia increased with patient age, with the highest probability of insomnia in patients who were ≥12 years of age when they received a TBI diagnosis. Polyphasic sleep and frequent awakenings are common and developmentally appropriate for infants; thus, an insomnia diagnosis is often not considered in infants and toddlers because of the inherent difficulty in doing so [45]. Similar to our findings, previous research has found that, in the general pediatric population, the prevalence of insomnia symptoms peaks between the ages of 11 and 12 [47,48]. Hormonal fluctuations associated with the onset of puberty may partially explain the increased prevalence of insomnia in adolescents [47,48]. Endocrine disruption and hormone deficiencies are common outcomes of TBI [7,49]. In our study, we also found that the diagnosis ‘difficulty sleeping’ peaked between the ages 7 and 9 years of age. We recently reported that pediatric patients with TBI who were diagnosed with a hypothalamic-pituitary disorder showed the highest prevalence in ages 7–11 [7,50]. TBI-induced endocrine disruptions may exacerbate SWD, including symptoms of insomnia and difficulty sleeping. Additional work is necessary to understand the interplay between TBI-induced endocrine disruption and the development of SWD in the pediatric population.

We identified that patient age at initial TBI diagnosis, injury severity, and number of TBIs were all predictive of the time to onset for an SWD diagnosis. Here, we report the time from brain injury to SWD with a median time to diagnosis of approximately 2 months post-TBI. Our data also indicated that older patients had a shorter time to SWD diagnosis. We found that the ‘other’ category had a high prevalence that decreased with patient age. Sleep disturbances are often difficult to assess in the pediatric population, and older patients may have had a shorter time to SWD onset, and fewer non-specific ‘other’ diagnoses, because older patients could clearly communicate their symptoms to caregivers and clinicians. During early years, sleep assessments rely on the interpretation of children’s sleep by caregivers. As mentioned previously, polyphasic sleep and frequent awakenings are common for infants and young children and an SWD may be difficult for caregivers to discern. This challenge of recognizing and diagnosing SWD in infants and young children may also explain why our incidence of SWD after TBI did not reflect the incidence of pediatric TBI in the general population which has its first peak in the first two years of life. Additionally, patients with severe TBI had the shortest onset to SWD compared to mild and moderate injuries. We hypothesize that this short onset to SWD diagnosis is likely due to the severity of the sleep disturbances. Children with moderate-to-severe TBI demonstrated the highest scores for sleep disturbance at 24 months post-injury compared to mild TBI and controls [42].

Interestingly, we found that the more TBIs a patient sustained, the longer it took for a subsequent SWD diagnosis from the time of their initial TBI diagnosis. In a pediatric TBI study, 77% of patients presented with at least one concurrent injury that involved the face, head, thorax, and lower or upper extremities [51]. A pediatric patient who has sustained multiple TBIs is likely to have polytrauma and life-debilitating physical symptoms that may necessitate long-term medical treatment prior to the diagnosis and management of sleep disturbances. Furthermore, patients with a single TBI may have an increased recognition of post-injury impairments, such as sleep disturbances, and are therefore more likely to seek medical attention for SWD as an injury complication.

Our group recently performed a retrospective review of patients at Phoenix Children’s Hospital, the site of the current study, and reported clinical indications and risk factors associated with pediatric abusive head trauma [52]. Based on our previous work on abusive head trauma and the epidemiology of brain injuries concurrent with domestic violence [52,53,54], we postulate that there are pediatric patients in the current study who sustained abusive head traumas as a mechanism of injury. If a patient has inflicted or non-accidental head trauma perpetrated by a caregiver or family member, it is less likely that the patient will receive medical attention for a subsequent TBI-induced morbidity such as SWD. For example, in an extended follow-up of neurological outcomes after severe abusive head trauma in children, almost one-fifth of patients did not receive any form of rehabilitation therapy after their injury [55]. This lack of follow-up care may explain why patients in our study who sustained multiple TBIs had a longer onset to subsequent SWD diagnoses. Our findings differ from previous works that report that athletes with multiple concussions report more sleep difficulties and worse sleep quality than controls [56,57]. However, studies focused on sports-related concussions exclude toddlers and infants, which are at a higher risk for abusive head trauma [52,53].

Mechanical forces applied to the head or brain initiate TBI, wherein sequential pathophysiological processes permanently change neurological function, evolving into a disease [58,59]. Although the pathophysiology of TBI-induced SWD is not completely clear, clinical and translational studies attribute these sleep disturbances to structural damage to the sleep circuitry of the brain, circadian rhythm disorders, pain, hormonal dysfunction, and an activation of the immune system [13,60,61,62,63]. TBI causes neuronal cell death, ischemia, hemorrhage, and disruption of the blood–brain barrier. These primary insults initiate a secondary injury response of cellular and molecular cascades, including central inflammation and the production of cytokines [64]. Cytokines not only communicate with other immune cells, but they are also key messengers in immune-brain interactions, including sleep [65,66,67,68]. Extensive research demonstrates that pro-inflammatory cytokines, such as interleukin-6 (IL6), IL1β, and tumor necrosis factor-α (TNF), are also pro-somnogenic [67,69]. Using a mouse model, we previously demonstrated that altered sleep after a diffuse TBI was temporally associated with inflammation and elevated cytokines [62,70,71]. Chronic inflammation and elevated cytokines have been reported at 3 months post-TBI and predicted unfavorable outcomes at 6- and 12 months post-injury [72]. Altered pro-inflammatory cytokines have also been reported in patients with SWD, such as insomnia [70]. Together, these studies suggest that inflammation may be a modifiable target to treat SWD after pediatric TBI, which warrants further research in this area [13].

Although clinical and preclinical data indicate that TBI can lead to an SWD, other events associated with the TBI that we could not account for may have contributed to SWD in this pediatric population. For instance, TBI-induced disabilities and persisting symptoms can place stress on the patients and their caregivers [71]. These chronic morbidities require ongoing medical visits, rehabilitation, and emotional, social and educational support [71]. Stress from the hospital stay, as well as stress from ongoing functional deficits, can lead to SWD in children and is a plausible inciting factor of SWD following a TBI [73,74,75,76]. Parental stress and maladaptive behavior by a parent or caregiver in response to trauma can also influence sleep in children [77]. It should also be noted that pain following a TBI is prevalent and the co-occurrence of stress and pain may alter sleep in the pediatric population [78,79]. Stress in response to trauma coupled with ongoing functional deficits can also lead to psychological symptoms, and it is well-recognized that SWDs frequently accompany mental health diagnoses [80,81]. Together, these symptoms that occur secondary to a TBI can influence sleep beyond the structural damage caused by the initial mechanical injury and should be considered when assessing SWD subsequent to TBI diagnoses in pediatric patients.

The results of this study should be interpreted in the context of several limitations that are primarily due to the retrospective nature of the study and the use of only ICD-9/10-CM codes for diagnosis. Reviewing prior documentation for billing codes eliminates much of the clinical scenario surrounding each diagnosis, including structural changes caused by TBI which cannot be included in the analyses. We could not determine whether the diagnosis date of SWD or TBI was the actual onset date of the disease or if there was a delay from disease onset to treatment. Another limitation included the diversity of sleep-related codes included in the ICD-9-CM and ICD-10-CM; the variety and infrequency of each individual code required the use of an ‘other’ category that combined several SWD codes. As such, ICD codes do not always provide sufficient clinical specificity to describe the complexity of SWD that can occur in pediatric TBI. Our retrospective study only captured data on patients seeking medical treatment for their SWD at a single institution. Patients who sought treatment for their SWD with a primary care physician outside our institution was not included in this study, and likely contributed to the low prevalence observed in our dataset. Our reported prevalence of SWD is lower than previously published longitudinal studies in pediatric TBI patients [29].

Another related issue that highlights the limitation of ICD codes is that sleep in adolescents is influenced by a variety of factors [82,83,84], including but not limited to the pubertal transition [85,86,87,88]. Indeed, studies investigating the effects of hormones and hormonal fluctuations on sleep show that changes to hormone release can affect sleep in children [88,89,90,91]. Sleep is a complex process and an accurate SWD diagnosis often requires the patient or the patient’s guardians to have an awareness of both normal sleep patterns and abnormal sleep. For the current dataset, the issue would most likely manifest as an underestimation of SWD in children, given that children and caregivers might view SWD during puberty as normal and not seek care. Our retrospective study design did not have matched controls and objective measures of SWD. A prospective study and long-term follow up of the effects of TBI in children on SWD is warranted.

## 5. Conclusions

Current research indicates that proper sleep may be critical for recovery from TBI, and chronic disturbances in sleep impact recovery and quality of life [13,14,92]. Our findings help to address a deficit in the research by observing the frequency and type of SWD in children after TBI and to characterize these disturbances based upon factors such as age, severity of TBI, and multiple insults through a retrospective analysis of patients with TBI and subsequent treatment for SWD. After pediatric TBI, sleep regulation may be impaired, and children are especially vulnerable to impairments in recovery due to SWD [12,40]. Thus, it is important for pediatric clinicians to be aware of TBI-induced SWD and to provide early intervention that aids recovery. In conclusion, we were able to identify associations between TBI characteristics and subsequent SWD diagnosis in the pediatric population. Although there are limitations to this retrospective study, the estimated relationships represent an important step toward increasing awareness and education among providers, parents, and patients about SWD following TBI. Future prospective studies would likely be able to address many of the limitations described above and may provide an even greater understanding of the relationships between TBI and SWD.

## Figures and Tables

**Figure 2 biology-11-00600-f002:**
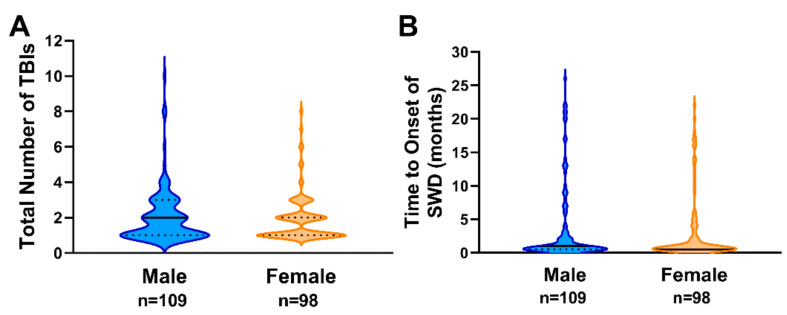
(**A**) Distribution of the total number of TBIs prior to SWD by gender. (**B**) Distribution of time from TBI to onset of SWD by gender. Medians and quartiles are represented by solid black lines and dashed black lines, respectively.

**Figure 3 biology-11-00600-f003:**
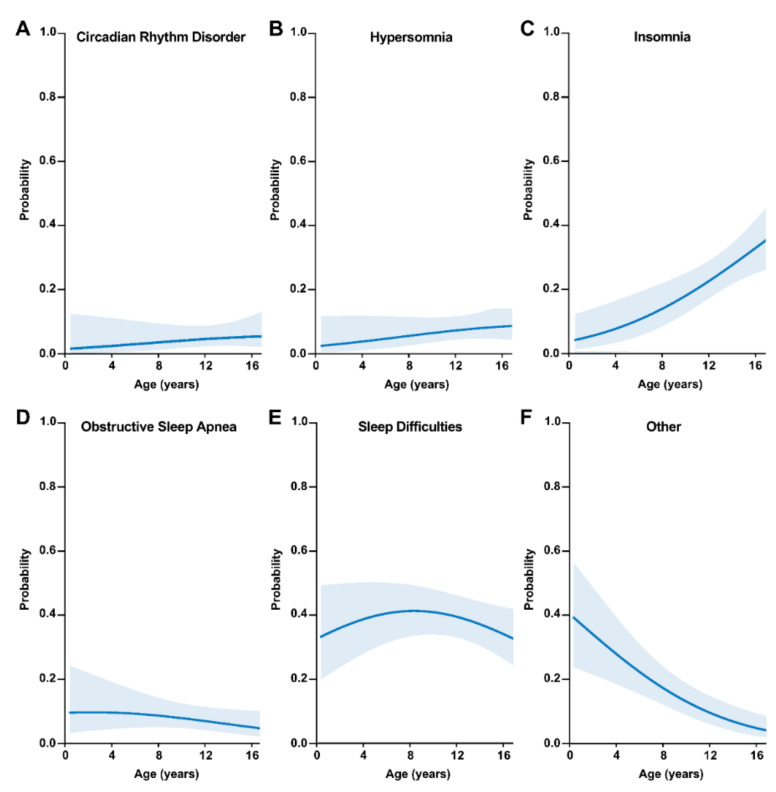
Predicted probability point estimates (solid blue lines) and corresponding 95% confidence intervals (blue shaded areas) for the types of sleep-wake disturbances predicted by patient age at initial TBI, from the top-ranked multinomial logit model. Sleep-wake disturbances included in the model were (**A**) circadian rhythm disorder, (**B**) hypersomnia, (**C**) insomnia, (**D**) obstructive sleep apnea, (**E**) sleep difficulties, and (**F**) other.

**Figure 4 biology-11-00600-f004:**
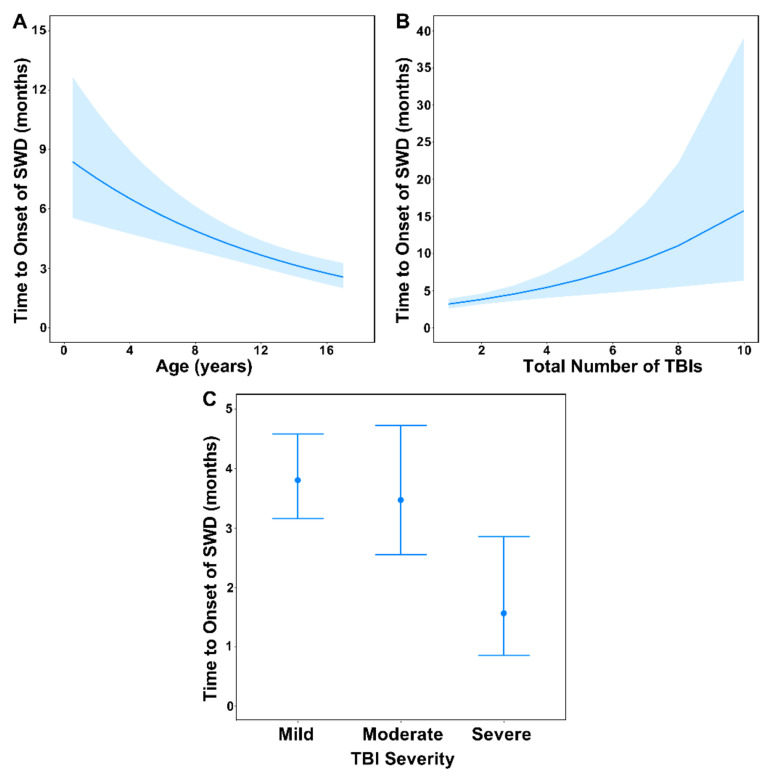
Marginal effects point estimates (solid blue lines or blue circles) and corresponding 95% confidence intervals (blue shaded areas or error bars) from the top-ranked negative-binomial generalized linear model predicting the effects of (**A**) age at initial TBI, (**B**) total number of TBIs prior to the onset of sleep-wake disturbances, and (**C**) severity of initial TBI on the time to onset of sleep-wake disturbances following injury.

**Table 1 biology-11-00600-t001:** The reduced categories of sleep-wake disturbances (SWD) after TBI. Sample sizes are provided in parentheses.

Type of SWD	Males	Females	Total
circadian rhythm sleep disorder	5.5% (6)	3.1% (3)	4.4% (9)
hypersomnia/excessive daytime sleepiness	6.4% (7)	9.2% (9)	7.7% (16)
insomnia	22.9% (25)	28.6% (28)	25.6% (53)
obstructive sleep apnea (OSA)	10.1% (11)	5.1% (5)	7.7% (16)
sleep difficulties/sleep disorders	39.5% (43)	41.8% (41)	40.6% (84)
other	15.6% (17)	12.2% (12)	14.0% (29)

**Table 2 biology-11-00600-t002:** Information-theoretic model selection results of all considered multinomial logit models for predicting the types of sleep-wake disturbance (Type). Predictor variables were the age in years at initial TBI (Age), Gender, severity of initial TBI (Severity), the total number of TBIs prior to the onset of sleep-wake disturbances (TBIs), and the time (months) to onset of sleep-wake disturbances (Onset).

Model	*K* ^a^	Log-Lik ^b^	AIC*_c_* ^c^	ΔAIC*_c_* ^d^	Wt ^e^
Type ~ Age	10	−299.52	620.16	0.00	0.55
Type ~ Age + Onset	15	−294.99	622.49	2.33	0.17
Type ~ Age + TBIs	15	−295.09	622.68	2.52	0.16
Type ~ Age + Onset + TBIs	20	−289.56	623.64	3.48	0.10
Type ~ Age + Gender	15	−297.77	628.05	7.89	0.01
Type ~ Age + Gender + TBIs	20	−293.24	630.99	10.83	0.00
Type ~ Age + Gender + Onset	20	−293.42	631.35	11.19	0.00
Type ~ Age + Severity	20	−293.58	631.68	11.52	0.00
Type ~ Age + Gender + Onset + TBIs	25	−287.63	632.45	12.29	0.00
Type ~ Age + Severity + Onset	25	−289.23	635.63	15.47	0.00
Type ~ Age + Severity + TBIs	25	−290.17	637.53	17.37	0.00
Type ~ Onset	10	−308.93	638.98	18.82	0.00
Type ~ Onset + TBIs	15	−303.91	640.32	20.16	0.00
Type ~ 1	5	−315.11	640.52	20.36	0.00
Type ~ Age + Severity + Onset + TBIs	30	−285.09	640.75	20.59	0.00
Type ~ Age + Severity + Gender	25	−291.88	640.94	20.78	0.00
Type ~ TBIs	10	−311.00	643.12	22.96	0.00
Type ~ Age + Severity + Gender + Onset	30	−287.74	646.04	25.88	0.00
Type ~ Gender + Onset	15	−307.14	646.80	26.64	0.00
Type ~ Age + Severity + Gender + TBIs	30	−288.29	647.15	26.99	0.00
Type ~ Gender	10	−313.07	647.26	27.10	0.00
Type ~ Severity + Onset	20	−301.42	647.36	27.20	0.00
Type ~ Severity	15	−307.61	647.74	27.58	0.00
Type ~ Gender + Onset + TBIs	20	−301.78	648.08	27.92	0.00
Type ~ Gender + TBIs	15	−308.90	650.31	30.15	0.00
Type ~ Age + Severity + Gender + Onset + TBIs	35	−283.22	651.17	31.01	0.00
Type ~ Severity + Onset + TBIs	25	−297.41	651.99	31.83	0.00
Type ~ Severity + TBIs	20	−304.52	653.55	33.39	0.00
Type ~ Severity + Gender	20	−305.70	655.91	35.75	0.00
Type ~ Severity + Gender + Onset	25	−299.83	656.84	36.68	0.00
Type ~ Severity + Gender + Onset + TBIs	30	−295.45	661.47	41.31	0.00
Type ~ Severity + Gender + TBIs	25	−302.49	672.16	52.00	0.00

^a^ Number of model parameters. ^b^ log-likelihood of model. ^c^ Akaike’s Information Criterion (AIC) corrected for small sample size. ^d^ Difference between AIC*_c_* of model and AIC*_c_* of top-ranked model. ^e^ Model weight.

**Table 3 biology-11-00600-t003:** Information-theoretic model selection results of all considered negative-binomial generalized linear models for predicting the time (months) to onset of sleep-wake disturbances (Onset). Predictor variables were the age in years at initial TBI (Age), Gender, severity of initial TBI (Severity), and the total number of TBIs prior to the onset of sleep-wake disturbances (TBIs).

Model	*K* ^a^	Log-Lik ^b^	AIC*_c_* ^c^	ΔAIC*_c_* ^d^	Wt ^e^
Onset ~ Age + Severity + TBIs	6	−492.15	996.70	0.00	0.50
Onset ~ Age + Severity + Gender + TBIs	7	−491.92	998.37	1.67	0.22
Onset ~ Age + TBIs	4	−495.33	998.83	2.13	0.17
Onset ~ Age + Gender + TBIs	5	−495.15	1000.57	3.87	0.07
Onset ~ Age	3	−498.61	1003.31	6.61	0.02
Onset ~ Age + Gender	4	−498.12	1004.42	7.72	0.01
Onset ~ Age + Severity	5	−498.07	1006.42	9.72	0.00
Onset ~ Age + Severity + Gender	6	−497.54	1007.47	10.77	0.00
Onset ~ TBIs	3	−502.17	1010.43	13.73	0.00
Onset ~ Severity + TBIs	5	−500.77	1011.80	15.10	0.00
Onset ~ Gender + TBIs	4	−501.99	1012.16	15.46	0.00
Onset ~ Severity + Gender + TBIs	6	−500.55	1013.49	16.79	0.00
Onset ~ 1	2	−505.54	1015.11	18.41	0.00
Onset ~ Gender	3	−505.07	1016.23	19.53	0.00
Onset ~ Severity	4	−505.50	1019.16	22.46	0.00
Onset ~ Severity + Gender	5	−505.01	1020.29	23.59	0.00

^a^ Number of model parameters. ^b^ log-likelihood of model. ^c^ Akaike’s Information Criterion (AIC) corrected for small sample size. ^d^ Difference between AIC*_c_* of model and AIC*_c_* of top-ranked model. ^e^ Model weight.

## Data Availability

Patient data are protected and were made available to authors through an approved IRB protocol.

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
