# Peer review of "Traumatic Brain Injury Characteristics Predictive of Subsequent Sleep-Wake Disturbances in Pediatric Patients"

_biology, 2022, doi:10.3390/biology11040600_

Round 1

Reviewer 1 Report

This manuscript is about an important and often neglected problem. It is well written. I have some minor comments and some more essential ones.

Minor comments:

  • I wonder whether the 'Simple summary' is really simple. If it is meant for lay audience, it probably should be rewritten.
  • Figures 1 and 2 differ in the way the legends are composed. In Figure 2, the elements are been divided by a period, in figure 1 by a comma (partly wrongly placed). Choose one style (preferably: a period).
  • Line 256: the authors state that research on sleep disturbances in children following TBI is limited. However, in line 51 they stated that "an increasing body of literature has revealed....". I suggest to say nothing about the amount of research. That topic has not been investigated in this study.
  • Line 279: I assume, a period is missing.
  • Line 386: the statement that 'After pediatric TBI, sleep regulation is impaired...' seems to be to firmly. If only 4% of the studied population can be found with SWD, one can only say that 'sleep regulation MAYBE impaired'.

Essential comments

  • In line 109/110 the authors present the number of patients found with SWD: 207. That is about 4% of the total population. This is in sharp contrast with the statement in line 53, where they state: "that disturbed sleep is one of the most frequently reported post-injury symptoms".  It takes until line 365 that the authors comment on this difference. Reading the manuscript, I was wondering from line 109 what the differences in prevalence were compared to other studies. Please, add the prevalence numbers out of other studies, and reflect on it in the results, not only in the limitations part.
  • Line 119: the authors describe that they combined categories form the ICD's in order to make it possible to do the statistics. I was interested what categories were combined, but couldn't find it. Please add a table in which you show the ICD-categories that have been combined into the 5 new categories.
  • line 258: the authors state that 'Our results provide evidence that pediatric TBI can lead to the development of SWD'. This is incorrect. The study was about the development of SWD after TBI (and prognostic factors), not because of TBI. Other causes are quit well conceivable (stress after hospital admission; stress because of function loss; environmental influences....).
  • Furthermore, from line 335-353, the authors hypothesize about the possible causes of the SWD in terms of inflammation and structural damages. In this part of the discussion, other possible causes (see former bullet) should be incorporated, including parental traumatic stress. There are a lot of pyschological (cognitve, emotional, social) issues after TBI that can play a (major) role in developing behavioural problems. These factors must be incorporated this into the manuscript, with proper references.

Reviewer 2 Report

The manuscript is well written, materials and methods well described and the statistical analysis sound. The results are well presented and graphically supported when necessary.

Comments:

Nevertheless some critical points remain which are addressed by the authors themselves in the discussion section. As in all of these kind of studies that are based on IDC-codes, inexact or even false coding cannot be excluded, especially as the codes are also used for billing. Also a high number of included patients may smoothen this problem it cannot be neglected and inexact data do not get better by an elaborate statistical analysis.

What is the meaning of the excursus on the GCS in the introduction? It might be better placed in the discussion section with regard the ICD-coding. Was the GCS used in the patient files for ICD-Coding or another score?

One of the major problems of these kind of studies is the fact that structural changes of TBI as found on imaging cannot be considered in the analysis.

Why were patients with multiple TBIs not excluded as it remains unclear, how and or which of the multiple TBIs has influenced the SWD. Was the first TBI the index TBI for calculation for the time until SWD or the last TBI? In neither case the results are reliable.

The distribution of the patients‘ age in this study does not reflect the incidence of paediatric TBI in the general population, which has a first peak in the first two years of life.

The lists of the ICD-codes are added as supplementary file. It is not possible for the reader to extract the 12 SWD which were reduced to six for the study. Maybe this can be marked somehow in the appendix.

There are two lists of ICD-10 codes for TBI with different coding numbers? Explanation?

Round 2

Reviewer 1 Report

I thank the authors for their appropriate answers to my comments.

Author Response

We thank you for your constructive comments that improved the manuscript.